# Analysis of Knowledge, Attitudes and Behaviours of Health Care Workers towards Vaccine-Preventable Diseases and Recommended Vaccinations: An Observational Study in a Teaching Hospital

**DOI:** 10.3390/vaccines11010196

**Published:** 2023-01-16

**Authors:** Marianna Riccio, Mattia Marte, Valentin Imeshtari, Francesca Vezza, Vanessa India Barletta, David Shaholli, Corrado Colaprico, Maria Di Chiara, Elena Caresta, Gianluca Terrin, Paola Papoff, Giuseppe La Torre

**Affiliations:** 1Department of Public Health and Infectious Diseases, Sapienza University of Rome, Piazzale Aldo Moro 5, 00185 Rome, Italy; 2Department of Maternal and Child Health, Policlinico Umberto I, Sapienza University of Rome, 00185 Rome, Italy; 3Pediatric Intensive Care Unit, Department of Pediatrics, Umberto I Policlinico, Sapienza University of Rome, 00185 Rome, Italy

**Keywords:** healthcare workers, recommendations, vaccination, knowledge, attitudes, behaviour

## Abstract

Background: Recommended vaccinations are the cheapest and most effective measure to reduce the risk of transmission and related complications, especially in high-risk healthcare settings. This study aimed to evaluate the knowledge, attitudes and behaviours of HCWs in relation to national recommendations. Methods: A transversal study was conducted through administration of a questionnaire by personal interview. The following care units were involved: Paediatric, Neonatal, Cardiac Surgery and General Intensive Care Units and Infectious Diseases Unit. Results: The study sample comprised 308 HCWs. Half the sample were aware of the vaccination recommendations, with occupation and age found to be predictive factors (OR = 9.38, 95%CI: 2.07–42.41; OR = 0.36, 95%CI: 0.22–0.60). A higher percentage defined the diseases as a risk for their patients’ health, although this perception was lower in the over-40 age group. In several cases, there were statistically significant differences between the care units (*p* < 0.001). Around three-quarters of the sample agreed that vaccination should be mandatory; willingness to undergo a future booster vaccination was statistically correlated with the variables of age and care unit (*p* < 0.001, *p* = 0.03). Conclusion: The protection of health in the workplace can be achieved through some strategic actions, such as the implementation of educational strategies, and protocols for the monitoring of immunocompetence and the improvement of vaccination.

## 1. Introduction

Healthcare workers (HCWs) have a greater risk of contracting vaccine-preventable diseases (VPDs) than the general population [1,2,3]. Some VPDs are highly contagious and can cause healthcare-associated infections with severe complications, especially in individuals with comorbidities or immune-depression, in new-borns and pregnant women [4,5]. Intensive care units are related to a higher number of infections, due to a higher chance of exposure and the presence of patients with a higher risk of complications [6,7,8]. Cases of infection within an intensive care unit can have significant clinical and organisational consequences: they can compromise unstable medical conditions or susceptible HCWs, and they can also cause economic, occupational, personal and family implications [4,9,10].

Some VPDs are often considered exclusive paediatric diseases, but recent epidemiological studies have shown that they can affect adults too. The average age of these infections has now shifted from the first to the third decade of life, with a more severe clinical course and a greater risk of complications in adults [6,10,11]. In addition, the vaccine coverage of HCWs is not completely satisfactory, compromising the aim of reducing or eliminating VPDs [1,10]. Thus, the vaccination of HCWs is one of the most important measures to control infections. The World Health Organization (WHO) has published specific recommendations to support national policies for the vaccination of HCWs against tuberculosis, hepatitis B, poliomyelitis, diphtheria, measles, rubella, meningococcal disease, influenza, chickenpox and pertussis [12]. The Italian National Preventive Vaccination Plan (PNPV) 2017–2019, which is still in force, issued specific vaccination recommendations for professional categories at risk of exposure [13]. The efficacy of vaccinations has led to a change in the epidemiological trend of infectious diseases and, consequently, to a diminished public perception of their severity and risk. These factors could explain actual growing doubts about safety and the need for vaccination programs. [14,15].

Several studies have investigated people’s knowledge, attitude and behaviour about vaccine recommendations, especially referred to parents/caregivers [16,17,18] and pregnant women [19,20,21]. Only recently surveys focused on HCWs’ practices and perceptions of vaccinations and their knowledge of the ones required for their role by the authorities. An adequate knowledge of the opinions of specific working classes is essential to develop targeted and effective safety measures for professional practice [22,23,24,25,26].

The aims of the present study are: to analyse knowledge, attitude and behaviour (KAB model) of HCWs in relation to measles, mumps, rubella, chickenpox, pertussis and meningococcal disease and their recommended vaccinations; to describe self-reported immunity to these diseases in terms of previous vaccination or natural infection; to identify the sociodemographic/professional variables associated with the outcomes of interest; and to identify potential practical and organizational solutions to improve the safety of healthcare facilities.

## 2. Materials and Methods

### 2.1. Study Design

A cross-sectional study, according to the Strengthening the Reporting of Observational Studies in Epidemiology (STROBE) statement, was carried out between December 2021 and August 2022.

### 2.2. Setting and Sample

This cross-sectional single-centre study was carried out at the teaching hospital Policlinico Umberto I of Rome. The inclusion criteria were care units with a high risk of exposure and complications. Therefore, HCWs from the Neonatal and Paediatric ICUs (NICU and PICU), the Cardiac Surgery ICU (CSICU), the General ICU and the Infectious and Tropical Disease Unit were considered. It included physicians, medical residents, nurses, paediatric nurses and nursing support staff. Participants were recruited during their work shifts and a questionnaire was administered.

Informed written consent was requested for voluntary participation in the study.

The study was conducted in compliance with the Declaration of Helsinki concerning ethical principles for the protection of human participants in medical research and the General Data Protection Regulation (GDPR) 2016/679 on privacy.

### 2.3. The Questionnaire

The questionnaire was administered by personal interviews during working hours and consisted of four sections. The first section was composed of nine questions regarding socio-demographic and occupational information. The other three sections investigated knowledge, attitudes, and behaviours and were composed respectively of six, five, and six closed-ended questions, with the possible answers of yes/no/cannot remember or, on a 3-point Likert scale, agree/not sure/disagree. It was developed on the basis of previously published surveys [22,23,24,25,26]. A pilot study among 30 HCWs was carried out for assessing the readability of the items and the reliability of the questionnaire. We reported very high reliability of the questionnaire (Cronbach alpha = 0.900).

The questions were developed following an investigation of the available scientific evidence, using the Knowledge-Attitude-Behaviour (KAB) theoretical approach. Information was collected on the following themes: information and knowledge of vaccine recommendations, perception of the risk, severity and transmissibility of the disease, belief in the efficacy of the recommended preventive action (in the case of vaccination), and vaccination or natural infection history. The following vaccine-preventable diseases were considered: measles, mumps, rubella, chickenpox, whooping cough and meningococcal disease.

### 2.4. Sample Size

The sample size of the study was calculated using EpiCalc 2000. The following parameters were taken into account:

Proportion of HCWs who knew all the vaccinations recommended for them:

14.00% [23]

Null hypothesis value: 9.00%

Significance: 0.05

Power: 80%.

The final sample size of the study was 290 HCWs.

### 2.5. Statistical Analysis

Statistical analysis was performed using mean, standard deviation (SD), median, and minimum and maximum values for quantitative variables. For qualitative variables, frequencies and percentages were computed. Student’s *t*-test or the Mann–Whitney U test was applied for two-group comparisons, and ANOVA and the Kruskal–Wallis test were used for comparisons of more than two groups. The Kolmogorov–Smirnov test was used to verify the normal distribution of quantitative variables. The Pearson’s correlation coefficient was computed to estimate the direct or indirect correlation between variables. Descriptive, univariable, and multivariable analysis were carried out.

The univariable analysis investigated correlations between the independent variables (age, gender, marital status, years of service, occupation, care unit) and the outcomes of interest, i.e., the positive answers (yes/agree) for each item. The statistical significance was set at a *p*-value < 0.05.

Multivariable analysis was conducted using a multiple logistic regression model, considering the positive responses (yes/agree) as the outcome variables. The independent variables were dichotomised as follows: occupation (physicians, nurses, residents vs. other), gender (female vs. male), care unit (paediatric, neonatal, cardiac surgery and general ICUs vs. infectious diseases), age group (over 40 vs. under 40), children (yes vs. no), years of service (over 10 vs. under 10), and employer (university hospital vs. private).

Univariate analyses were considered to test the inclusion of each explanatory variable in the final models. Effects with a *p*-value < 0.20 were included in the multivariate analysis model.

Results of the logistic regression models were presented as an Odds Ratio with a 95% confidence interval (95% CI).

All statistical analyses were performed using SPSS for Windows, release 27.0 (IBM, Armonk, NY, USA). The statistical significance was set at a *p*-value < 0.05.

## 3. Results

### 3.1. Study Population

The study involved 308 HCWs (Table 1), comprising nurses (58.1%), residents (17.2%), physicians (15.3) and other HCWs (nursing support staff) 9.4%. The total number of the people working in the considered CUs was 345. Most respondents were female (69.2%) and the predominant age group was 30–39 years. The most common length of service was 0–5 years (41.6%), with 64.4% reporting 0–5 years of service within their current unit. 30.2% of the sample worked in the General ICU, 24.4% in the Infectious Diseases Unit, 24% in the NICU, 13% in the PICU and 8.4% in the CSICU. Almost half of the respondents (47%) were unmarried; the mean number of children was 0.8 (SD 0.4).

### 3.2. Descriptive Analysis

The percentages of positive responses (agree, yes) are reported descriptively in Figure 1.

Approximately half of the respondents were aware of vaccine recommendations. The most recognised recommendation was for measles (56.2%), meningococcal disease (56.8%), the least indicated was for mumps (47.7%). No particular differences were seen in the tendency to consider the disease as more dangerous than any side effect of its vaccine. Almost half the sample approved of the information and briefings provided by the NHS, with 49.7%, for example, considering them sufficient in relation to measles.

The risk of nosocomial transmission was perceived as greatest for meningococcal disease (80.2%) and lowest for mumps (64.4%). Vaccines for the diseases analysed were considered safe and effective by most of the respondents (95.5% for measles, dropping to 90.9% for meningococcal disease). On average, 91.5% considered the various diseases to be a potential risk factor to the health of patients.

The highest agreement towards the mandatory vaccination was for meningococcal disease (81.5%), followed by chickenpox (77.6%), measles (76.6%), pertussis (76%), rubella (75.3%) and mumps (75%).

Most of the respondents had not been vaccinated against any of the diseases considered in the last 10 years, with a vaccination rate for all diseases of less than 20%, except meningococcal disease (25.5%). The main reasons indicated for the lack of vaccination were: “previous natural infection” and “no active vaccination campaign”, while the reasons given for undergoing vaccination were “to protect my health” and “to protect my family’s health”. On average, 83% of participants were favourable to a booster vaccination in the event of inadequate vaccine coverage.

The main sources of information indicated were: medical consultation (65.6%), colleagues (56.2%), institutional websites (52.9%), scientific societies (52.9%), and scientific journals (48.4%); the least used were mass media and social media.

### 3.3. Univariate Analysis

The results of the univariate (chi-squared) analysis are reported in the Appendix A. Each table shows the results for a specific disease. The first column of the table reports the number associated with each question (refer to the questionnaire or Figure 1).

HCWs aged 20–29 were more likely to indicate vaccination as recommended than other age groups (chickenpox 72.4% *p* < 0.001, measles 75% *p* < 0.001). In relation to occupation, knowledge was greatest in residents (mumps/rubella 75.5% *p* < 0.001).

Nursing support staff were more likely to believe some diseases affected children only (measles 34.5% *p* = 0.002, chickenpox 31% *p* = 0.027). Believing pertussis to be an exclusively childhood disease was significantly correlated with physicians (31.9% *p* = 0.002).

There were statistically significant differences (*p* < 0.001) for the item, “the disease is more dangerous that the side effects of its vaccine”, with the greatest agreement seen among physicians and residents.

The perceived risk of nosocomial transmission was significantly correlated with age group, occupation, years of service, care unit and children. The greatest agreement was seen in HCWs aged 20–29 (measles 88.2% *p* = 0.001, rubella 78.9% *p* < 0.001). The highest percentage was observed in PICU/NICU workers (chickenpox 95%/79.7% *p* = 0.029, mumps 85%/71.6% *p* = 0.008).

In relation to perception of the disease as a risk factor for patients’ health, statistically significant differences were seen for: occupation, with the greatest perception of risk in residents (pertussis 98.1% *p* = 0.028, measles 100% *p* = 0.014, rubella 100% *p* = 0.011); care unit, with the greatest perception in PICU/NICU workers (pertussis 100%/97.3%, *p* = 0.001, rubella 100%/95.9%, *p* = 0.004). The greatest agreement was seen in the youngest HCWs than those over 40 years old (chickenpox *p* = 0.008, measles *p* = 0.014).

The mandatory nature of vaccination for HCW was often correlated with age group: there was a gradual drop in positive answers with increasing age (mumps *p* = 0.002, rubella *p* = 0.004, chickenpox *p* = 0.027).

Vaccination in the last 10 years showed a statistically significant correlation with most of the analyzed variables. In particular, in relation to occupation, the residents were most likely to be vaccinated (measles 37.7% *p* = 0.002, chickenpox 30.3% *p* = 0.001, pertussis 41.5% *p* < 0.001, as well as the youngest respondents 20–29 years (measles 36.8% *p* < 0.001, chickenpox 34% *p* = 0.001, meningococcal disease 50% *p* < 0.001). HCWs of CSICU, in some cases, were less likely to be vaccinated than those of other care units (meningococcal disease *p* = 0.027, measles *p* = 0.02).

The possibility of a booster vaccine in the event of inadequate coverage was more accepted by the youngest HCWs (aged 20–29 measles 93.4% *p* < 0.001, pertussis 93.4% *p* < 0.001, meningococcal disease 97.4%, *p* < 0.001). Willingness to undergo a new booster vaccination was also correlated with the care unit, for example, with greatest willingness in the PICU/NICU for vaccine measles (*p* = 0.043).

### 3.4. Multivariate Analysis

Each table shows the results for a specific disease. The first column of the table reports the number associated with each question (refer to the questionnaire or Figure 1).

#### 3.4.1. Measles

In relation to age (Table 2), the over-40 age group category was significant in four situations: increasing age was associated with reduced knowledge of the recommendation (OR = 0.41, 95%CI: 0.24–0.71), reduced trust in the information provided by the NHS (OR = 0.31, 95%CI: 0.14–0.66), reduced acceptance of mandatory vaccination (OR = 0.38, 95%CI: 0.20–0.73) and reduced willingness to receive a booster vaccination if inadequately covered (OR = 0.24, 95%CI: 0.13–0.48). The same category was also predictive of considering the disease as exclusively affecting children (OR = 3.66, 95%CI: 1.73–7.75) and of having contracted the disease in the past (OR = 6.20, 95%CI: 3.61–10.64). In relation to occupation, the categories physicians, nurses and residents were significant predictors for several outcome variables. They showed a better knowledge of the vaccination recommendation (OR = 4.35, 95%CI: 1.56–12.10; OR = 2.51, 95%CI: 1.05–6.02; OR = 3.83, 95%CI: 1.32–11.15) and perception of the risk of nosocomial outbreaks (OR = 13.77, 95%CI: 2.39–79.32; OR = 9.17, 95%CI: 1.88–44.72; OR = 21.92, 95%CI: 3.74–128.40). Residents were most likely to have received the vaccine in the last 10 years (OR = 1.96, 95%CI: 0.96–4.01). Working in the General ICU was predictive of knowledge of the vaccination recommendation (OR = 1.77, 95%CI: 1.02–3.08), while working in the CSICU was predictive of acceptance of a possible future booster vaccination (OR = 0.23, 95%CI: 0.09–0.56). Working in the NICU was associated with a higher probability of perceiving measles as a risk for patients (OR = 8.75, 95%CI: 1.99–38.44) and considering vaccination of HCWs as a protective factor for patients (OR = 4.96, 95%CI: 1.10–22.42).

#### 3.4.2. Mumps

Being female was a predictive factor for knowledge of the vaccination recommendation (OR = 1.75, 95%CI: 1.04–2.92) and perceiving the disease as more dangerous than its vaccine (OR = 2.37, 95%CI: 1.24–4.54) (Table 3); however, it was correlated with a reduced tendency to evaluate the disease as a risk to personal health (OR = 0.46, 95%CI: 0.25–0.86). Increasing age was associated with reduced trust in the information provided by the NHS (OR = 0.39, 95%CI: 0.18–0.83), reduced the perception of the disease as a risk for patients’ health (OR = 0.40, 95%CI: 0.17–0.91), the possibility to accept future booster vaccination, if necessary (OR = 0.24, 95%CI: 0.12–0.45). Physicians, nurses and residents were more likely to consider mumps a risk for hospital-acquired infection (OR = 9.15, 95%CI: 1.72–48.69; OR = 7.74, 95%CI: 1.64–36.50; OR = 17.88, 95%CI: 3.33–96.01); while physicians and residents were more likely to perceive the disease as more dangerous than the undesirable effects of its vaccine physicians (OR = 5.19, 95%CI: 1.49–18.09; OR = 11.37, 95%CI: 2.55–50.60). Residents were more likely to be aware of the vaccine recommendation (OR = 3.19, 95%CI: 1.53–6.63). Working in the PICU was correlated with a greater probability of considering the disease as more dangerous than the undesirable effects of its vaccine (OR = 4.75, 95%CI: 1.06–21.20), while working in the NICU increased the probability of perceiving mumps as a risk for patients (OR = 8.47, 95%CI: 1.94–36.97) and of considering the vaccination of HCWs as a protective factor for patients (OR = 5.01, 95%CI: 1.08–23.17). However, working in the CSICU reduced the probability of considering vaccination as a protective factor for patients (OR = 0.27, 95%CI: 0.08–0.90), as well as that of willingness to undergo a future booster vaccination (OR = 0.25, 95%CI: 0.10–0.62).

#### 3.4.3. Rubella

Being female was a predictive factor for considering the disease as more dangerous than the undesirable effects of its vaccine (OR = 2.13, 95%CI:1.12–4.07) (Table 4). The over-40 age group reduced trust in the information provided by the NHS (OR = 0.37) and in the safety and efficacy of the vaccine (OR = 0.23, 95%CI:0.06–0.84); it also reduced the perception of rubella as a risk for patients (OR = 0.39, 95%CI:0.17–0.90) and acceptance of a possible future booster vaccination, if necessary (OR = 0.22, 95%CI:0.11–0.45). The categories physician, nurse and resident had a better perception of the risk of hospital-acquired infection (OR = 6.46, 95%CI:1.45–28.71; OR = 5.15, 95%CI:1.35–19.67; OR = 11.49, 95%CI:2.55–51.84). Knowledge of vaccine recommendations and perception of the dangerousness of the disease was higher for physicians (OR = 6.81, 95%CI:1.70–27.20/OR = 5.01, 95%CI:1.45–17.37) and residents (OR = 9.38, 95%CI:2.07–42.41/OR = 10.71, 95%CI:2.41–47.61). Working in the PICU increased the probability of considering the disease as more dangerous than the undesirable effects of the vaccine (OR = 4.64, 95%CI:1.05–20.59) and of perceiving the risk of hospital-acquired infection (OR = 3.08, 95%CI:1.14–8.33); working in the NICU increased the probability of perceiving rubella as a risk to personal health (OR = 4.88, 95%CI:1.41–16.85); working in the CSICU showed a negative correlation with considering the vaccination of HCWs as a protective factor for patients (OR = 0.26, 95%CI:0.08–0.88). Having children reduced the tendency to consider the disease as affecting children only (OR = 0.36, 95%CI:0.15–0.83) and improved the acceptance of mandatory vaccination for nursing support staff (OR = 2.43, 95%CI:1.21–4.87).

#### 3.4.4. Chickenpox

Knowledge of the vaccine recommendation diminished in the over-40 age group category (OR = 0.44, 95%CI:0.27–0.74) (Table 5). This category also had less trust in the information provided by the NHS (OR = 0.49, 95%CI:0.29–0.80) and in the safety and efficacy of the vaccine (OR = 0.34). In relation to occupation, physicians, nurses and residents were more likely to recognise the risk of hospital-acquired infection (OR = 10.78, 95%CI:1.88–61.82; OR = 10.45, 95%CI:2.11–51.74; OR = 20.58, 95%CI:3.47–121.97) and to accept mandatory vaccination (OR = 5.08, 95%CI:1.06–24.37; OR = 5.34, 95%CI:1.27–22.49; OR = 6.52, 95%CI:1.36–31.32). The category residents was also predictive in other cases: perception of the disease as more dangerous than the undesirable effects of the vaccine (OR = 4.43, 95%CI:1.26–15.63) and vaccination in the previous 10 years (OR = 5.67, 95%CI:1.79–17.96). Working in the PICU increased the probability of perceiving chickenpox as more dangerous than the undesirable effects of its vaccine (OR = 4.92, 95%CI:1.14–21.35) and of recognising the risk of hospital-acquired infection (OR = 4.55, 95%CI:1.04–19.93). Working in the NICU increased the tendency to consider chickenpox as a risk for patients (OR = 5000.48, 95%CI:1.58–18.96). The CSICU was predictive of three items: vaccination of HCWs as a protective factor for patients (OR = 0.26, 95%CI:0.08–0.88), acceptance of mandatory vaccination (OR = 0.41, 95%CI:0.18–0.97) and acceptance of a booster vaccination in the event of inadequate coverage (OR = 0.17, 95%CI:0.07–0.45). HCWs with over 10 years of service as a predictive factor for some outcome variables: it increased the probability to consider chickenpox as a risk for patients’ health (OR = 0.32, 95%CI:0.19–0.55) and the probability of having been vaccinated in the last 10 years (OR = 0.17, 95%CI:0.07–0.45).

#### 3.4.5. Pertussis

Women were more likely to consider pertussis as dangerous (OR = 2.32, 95%CI:1.23–4.39), but less likely to consider vaccination of HCWs as a protective factor for patients (OR = 0.25, 95%CI:0.07–0.88) (Table 6). The over-40 age group category was predictive of three items: trust in the information provided by the NHS (OR = 0.44, 95%CI:0.21–0.93), perception of the disease as a risk for patients (OR = 0.38, 95%CI:0.16–0.90) and history of natural infection (OR = 2.86, 95%CI:1.62–5.04). Physicians and residents were more likely to be aware of the vaccine recommendation (OR = 3.02, 95%CI:1.09–8.37, OR = 7.02, 95%CI:2.44–20.20) and to perceive the disease as more dangerous than the undesirable effects of its vaccine (OR = 8.51, 95%CI:1.95–37.07; OR = 8.68, 95%CI:2.02–37.23). Being a physician, resident or nurse was also predictive of awareness of the risk of hospital-acquired infection (OR = 7.12, 95%CI:1.57–32.19; OR = 4.60, 95%CI:1.19–17.77; OR = 12.75, 95%CI:2.79–58.35). However, physicians and nurses were also more likely to consider pertussis as an exclusively childhood disease (OR = 2.73, 95%CI:1.15–6.50; OR = 2.39, 95%CI:1.19–4.77). Residents were most likely to have been vaccinated in the last 10 years (OR = 3.12, 95%CI:1.50–6.47). In relation to the care unit, PICU workers were most likely to consider the disease as a source of hospital-acquired infection (OR = 4.62, 95%CI:1.32–16.12), and NICU workers were most likely to consider it a risk for patients (OR = 7.93, 95%CI:1.82–34.57). Working in the CSICU reduced the probability of considering vaccination of HCWs as a protective factor for patients (OR = 0.23, 95%CI:0.07–0.79) and of willingness to undergo vaccination, if necessary (OR = 0.19, 95%CI:0.08–0.50).

#### 3.4.6. Meningococcal Disease

Being female was correlated with increased knowledge of the vaccine recommendation (OR = 1.98, 95%CI:1.17–3.33) and perception of the dangerousness of the disease in comparison with the undesirable effect of its vaccine (OR = 2.72, 95%CI:1.33–5.54) (Table 7). The over-40 age group had less trust in the information provided by the NHS (OR = 0.42, 95%CI:0.20–0.87), were less likely to consider the disease as a risk for patients’ health (OR = 0.35, 95%CI:0.12–(−0.02)), and were less likely to indicate vaccination against meningococcal disease as recommended. Physicians, nurses and residents had a greater perception of the risk of hospital-acquired infection (OR = 6.31, 95%CI:2.01–19.80; OR = 4.68, 95%CI:2.01–10.90; OR = 5.26, 95%CI:1.84–15.07) and a greater trust in the safety and efficacy of the vaccine (OR = 14.29, 95%CI:1.64–124.23; OR = 2.78, 95%CI:1.04–7.43; OR = 17.32, 95%CI:1.10–150.17). These occupations were also more accepting of mandatory vaccination (OR = 7.08, 95%CI:1.39–36.00; OR = 6.73, 95%CI:1.55–29.19; OR = 6.94, 95%CI:1.40–34.39). Residents were the only category in this variable that had a greater perception of the disease as more dangerous than the undesirable effects of its vaccine (OR = 9.24, 95%CI:2.02–42.19). In relation to the care unit, General ICU and CSICU workers were predictive factors for having been vaccinated in the previous 10 years (OR = 0.51, 95%CI:0.27–0.96; OR = 0.12, 95%CI:0.02–0.62) and NICU workers were a predictive factor for considering meningococcal disease as a risk to personal health (OR = 2.50, 95%CI:1.04–5.99). Working in the CSICU was associated with a lower probability of accepting mandatory vaccination (OR = 0.38, 95%CI:0.16–0.92) and a possible future booster vaccination (OR = 0.22, 95%CI:0.09–0.54).

#### 3.4.7. Source of Information 

With increasing age, the use of scientific societies, scientific journals and institutional websites decreases (OR = 0.41 for all) (Table 8). Physicians, nurses and residents were correlated with a greatest probability of using scientific journals (OR = 88.81, 95%CI:18.97–415.72; OR = 5.46, 95%CI:1.57–19.00; OR = 7.73, 95%CI:1.99–30.04) and institutional websites (OR = 4.15, 95%CI: 1.42–12.10; OR = 3.58, 95%CI:1.53–8.36; OR = 7.75, 95%CI:1.81–33.27). Physicians and residents were less likely to indicate mass media (OR = 0.30, 95%CI:0.12–0.74; OR = 0.13, 95%CI:0.04–0.43) and social media (OR = 0.15, 95%CI:0.04–0.67; OR = 0.16, 95%CI:0.05–0.55). To be a nurse was a predictive factor for using medical consultation, such as source of information (OR = 3.98, 95%CI:2.08–7.65). In relation to care units, working in the CSICU and PICU was a predictive factor for using medical consultation (OR = 0.52, 95%CI:0.31–0.90; OR = 3.76, 95%CI:1.31–10.43) and colleagues (OR = 2.55, 95%CI:1.03–6.28; OR = 2.76, 95%CI: 1.28–5.92); working in General ICU was a predictive factor for using scientific journals (OR = 0.54, 95%CI:0.30–0.96).

## 4. Discussion

This study generally indicates a lack of knowledge about vaccination recommendations, an inadequate perception of the dangerousness and risk of the diseases, and a lack of trust in the information provided by the Italian National Health Service. While, on the whole, the respondents had a positive attitude towards vaccine efficacy and safety, they were less accepting of mandatory vaccinations and future booster vaccinations, if necessary.

Almost half the sample did not correctly indicate the Italian vaccination recommendations for HCWs. Poor knowledge might suggest a lack of interest in or attention towards the prevention and control of infections [27,28,29]. The generally poor knowledge of recommendations has also been reported in other recent surveys, both national [22,23,30] and international [25,31,32,33]. In some of these, unlike in the present study, younger HCWs were less knowledgeable [22,25,33], while others, like us, found a greater knowledge of the recommendations in younger HCWs [23,30]. As with our survey, in most of these studies physicians were better informed than other HCWs [22,23,25,33,34].

Around three-quarters of our participants supported a policy of mandatory vaccination for HCWs at risk of exposure. Several studies have addressed the same topic [9,14,26]. In some studies, the HCWs’ attitude differed depending on the type of patients being cared for and on the specific disease [35,36]. Mandatory vaccination is an effective measure to increase immunisation rates, but it is a controversial subject that introduces important ethical questions [27,37]. A closer look at the evidence brings up some interesting points: acceptance of mandatory vaccination may be vaccine-specific, and in some situations, the opinions expressed by HCWs do not reflect their actual behaviour. In fact, some studies found low vaccination rates despite the HCWs’ general approval of mandatory vaccination, revealing a discrepancy between opinion and practice [28,29]. In our study, physicians, residents and nurses were most likely to be in favour of mandatory vaccination, while the over-40 age group were least likely to be in favour. Similar results for age and occupation can also be found in other studies [9,26,36,37].

The present study also investigated the perception of the safety and efficacy of the vaccines for the diseases considered, and it was essentially positive. Trust in meningococcal and chickenpox vaccines was lower than for the other vaccines. Trust in vaccines tended to diminish in the over-40 age group and in the HCWs with lower education level. The majority of our sample agreed that vaccination of HCWs is a protective factor for patients. However, there was a difference between the various care units: e.g., agreement with vaccination against measles, mumps and pertussis was higher in the NICU that in the Infectious Diseases Unit. Other studies have also found that HCWs have a good perception of the protective value offered by vaccines [23,31].

Despite the positive results in relation to the efficacy, safety and utility of vaccines, around 20% of our sample did not believe the diseases in question to be more dangerous than an undesirable effect of their vaccine. This is probably due to how natural infections are perceived and how the risks to personal health from vaccine side effects are evaluated [27,38,39]. In relation to care unit, PICU workers were more likely than those in the Infectious Disease Unit to consider some diseases as more dangerous than the undesirable effects of their vaccine. Furthermore, physicians and residents showed a statistically significant higher agreement with this item than the other occupations.

The likelihood of taking any preventive action greatly depends on the perceived benefits. Willingness to get vaccinated is correlated with individual beliefs about vaccines as well as about the perceived risk of contracting the disease [40,41]. Around three-quarters of the sample believe that exposure to measles, mumps, rubella, chickenpox and pertussis is a risk for their own health; this percentage was higher for meningococcal disease. Other studies have also reported a difference in perception, and relatively greater attention, towards influenza, tuberculosis and hepatitis B in comparison with the aforementioned diseases [9,24,39]. The risk perception increased in our sample when considering patients’ health. It was higher in the NICU compared with the Infectious Disease Unit and in younger age groups. The correlation between these results and the HCWs’ behaviour was not analysed in the present study, but literature evidence shows that perception of risk is an important factor to investigate, as it is crucial for the acceptance of vaccination [42,43,44]. Similar studies have been conducted by other authors. Napolitano [22] found a low perception of the risk of contracting VPDs; in that study, there was a higher perception in the intermediate age group and a lower perception in the NICU than in the CSICU. In another study [45], less than half the respondents defined as potentially susceptible believed themselves to be at risk: the perceived risk in this sample was considerably lower in ICU workers than in emergency workers.

Intention to vaccinate was investigated by asking about willingness to undergo a booster vaccination for the diseases in question in the event of inadequate vaccination coverage. About 20% of respondents were unwilling or undecided. Willingness tended to be lower in the over-40 age group. There were statistically significant differences between the other care units in our sample, for example, with NICU and PICU workers most willing to undergo the various vaccinations. It is important to remember that vaccine hesitancy does not equal outright vaccine refusal, but instead involves various degrees of indecision [27]. The lack of adhesion of HCWs has been attributed in the literature to various barriers, from the absence of management support strategies to inadequate knowledge of the scope and underestimation of the risk [9,46,47].

Most respondents reported having contracted chickenpox and measles. Around 20% of our respondents reported having been vaccinated against diseases considered in the last 10 years. The highest vaccination level was for meningococcal disease (27%). Residents and nurses were more likely to have been vaccinated. There were also statistically significant differences in relation to age group. Occupation and age are often reported in the literature as predictive factors for vaccination coverage [26,36,37,48]. Protection of personal health was the main motivation for vaccination in our sample, as also reported in other studies on vaccine recommendations [29,31,49,50].

### Strengths and Limitations of the Study

The cross-sectional nature of this study limits the ability to establish causal relationships between the identified correlations. Other limitations include the single-centre sample, the voluntary nature of participation and the self-reported data (e.g., immunity, natural infection). The study does have some strengths. Administration of the questionnaire by direct interview enabled an easier recognition of the concerns and doubts of HCWs in relation to this specific topic; the interview was carried out using an approach aiming to minimise any social-desirability bias. Another strong point was the specific healthcare settings included.

## 5. Conclusions

Innovative personalised strategies are needed to improve the adhesion to vaccination recommendations. These strategies should include specific training activities. Training objectives must focus not only on theoretical aspects and knowledge, but also on practices and behaviours, individual and social factors, and environmental and organisational aspects. The lack of adequate vaccination coverage and of up-to-date information may have implications for the safety of healthcare facilities, as well as negative consequences for the role and image of healthcare professionals. All this points to the need to create safe working environments oriented towards prevention. The protection of health in the workplace can be achieved through the development of organisational protocols based on careful risk assessment, health monitoring activities, and the implementation of protective measures, including vaccination.

## Figures and Tables

**Figure 1 vaccines-11-00196-f001:**
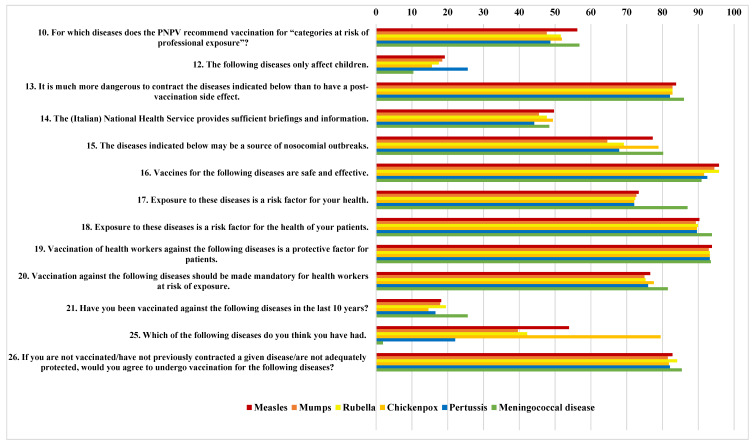
Descriptive analysis: percentages of positive responses (agree, yes) to the questionnaire.

**Table 1 vaccines-11-00196-t001:** Sample descriptive date.

Variable	N (%) or Average (DS)
Gender	
Female	213 (69.2)
Male	95 (30.8)
Age	
20–29	76 (24.7)
30–39	95 (30–8)
40–49	53 (17.2)
>50	84 (27.3)
Marital status	
Married/Cohabitant	131 (42.5)
Divorced	25 (8.1)
Unmarried	147 (47.7)
Widowed	5 (1.6)
Children N°	0.8 (0.4)
Occupation	
Physicians	47 (15.3)
Nurses	179 (58.1)
Residents	53 (17.2)
Other	29 (9.4)
Employer	
Public/University	270 (87.7)
Private	38 (12.3)
Care Unit	
Pediatric ICU	40 (13)
Neonatal ICU	74 (24)
Cardiac Surgery ICU	26 (8.4)
General ICU	93 (30.2)
Infectious Disease	75 (24.4)
Years of Service	
0–5	128 (41.6)
6–10	38 (12.3)
11–20	55 (17.9)
>20	87 (28.2)
Years of Service C.U.	
0–5	199 (64.6)
6–10	26 (8.4)
11–20	44 (14.3)
>20	39 (12.7)

**Table 2 vaccines-11-00196-t002:** Results of multivariate analysis—OR (95%CI) positive answers—measles.

Outcome Variable OR–CI 95%	Gender	Age	Marital Status ^	Occupation *			Care Unit **				Children	Years of Service	EMPL ***
	F	Over 40	Married/ Cohabitant	Physicians	Nurses	Residents	Gen. ICU	Cardiac	Neon.	Ped.	Yes	Over 10	Public/Univ
10_yes		0.41 (0.24–0.71)		4.35 (1.56–12.10)	2.51 (1.05–6.02)	3.83 (1.32–11.15)	1.77 (1.02–3.08)						
12_agree		3.66 (1.73–7.75)		0.17 (0.05–0.61)					0.40 (0.18–0.91)		0.52 (0.25–1.11)		
13_agree	2.29 (1.18–4.46)	0.42 (0.156–1.128)		18.02 (2.35–138.36)		8.52 (1.89–38.45)				4.03 (0.89–18.23)		2.79 (1.04–7.50)	
14_agree		0.31 (0.14 – 0.66)										2.28 (1.06–4.89)	
15_agree				13.77 (2.39–79.32)	9.17 (1.88–44.72)	21.92 (3.74–128.40)		0.44 (0.19–1.06)		3.07 (0.89–10.59)			0.16 (0.03–0.86)
16_agree	0.26 (0.05–1.32)			9.20 (0.94–90.02)	5.28 (1.34–20.78)								
17_agree				2.16 (0.94–4.97)						2.60 (0.96–6.99)		0.36 (0.21–0.61)	
18_agree									8.75 (1.99–38.44)			0.33 (0.14–0.79)	
19_agree	0.28 (0.08–1.01)								4.96 (1.10–22.42)				
20_agree		0.38 (0.20–0.73).						0.45 (0.19–1.06)			1.78 (0.92–3.45)		
21_yes						1.96 (0.96–4.01)						0.11 (0.04–0.28)	
25_yes		6.20 (3.61–10.64)		2.74 (1.15–6.55)	1.91 (1.05–3.48)					0.40 (0.18–0.88)			
26_agree		0.24 (0.13–0.48)						0.23 (0.09–0.56)					

^ reference: unmarried; * reference: Infectious Disease; ** reference: other health workers; *** reference: private.

**Table 3 vaccines-11-00196-t003:** Results of multivariate analysis—OR (95%CI) positive answers—mumps.

Outcome Variable OR–CI 95%	Gender	Age	Marital Status ^	Occupation *			Care Unit **				Children	Years of Service	EMPL ***
	F	Over 40	Married/ Cohabitant	Physicians	Nurses	Residents	Gen. ICU	Cardiac	Neon.	Ped.	Yes	Over 10	Public/Univ
10_yes	1.75 (1.04–2.92)					3.19 (1.53–6.63)						0.59 (0.35–0.97)	
12_agree		2.22 (1.09–4.55)	2.16 (1.10–4.28)	0.35 (0.13–0.97)							0.44 (0.20–0.98)		
13_agree	2.37 (1.24–4.54)			5.19 (1.49–18.09)		11.37 (2.55–50.60)				4.75 (1.06–21.20)		1.95 (1.02–3.73)	
14_agree		0.39 (0.18–0.83)										2.34 (1.11–4.95)	
15_agree				9.15 (1.72–48.69)	7.74 (1.64–36.50)	17.88 (3.33–96.01)	0.54 (0.32–0.92)			2.36 (0.91–6.12)			0.15 (0.03–0.73)
16_agree						3.48 (0.45–26.84)							
17_agree	0.46 (0.25–0.86)		1.61 (0.91–2.84)							3.35 (1.22–9.16)		0.33 (0.19–0.58)	
18_agree		0.40 (0.17–0.91)		8.35 (1.09–64.22)					8.47 (1.94–36.97)				
19_agree	0.30 (0.09–0.95)				2.92 (1.06–8.05)			0.27 (0.08–0.90)	5.01 (1.08–23.17)				
20_agree		0.47 (0.20–1.08)									2.74 (1.35–5.56)	0.37 (0.15–0.91)	
21_yes	0.53 (0.26–1.08)				2.78 (1.06–7.27)	4.59 (1.61–13.10)						0.10 (0.04–0.28)	
25_yes												4.58 (2.81–7.48)	
26_agree		0.24 (0.12–0.45)						0.25 (0.10–0.62)					

^ reference: unmarried; * reference: Infectious Disease; ** reference: other health workers; *** reference: private.

**Table 4 vaccines-11-00196-t004:** Results of multivariate analysis—OR (95%CI) positive answers—rubella.

Outcome Variable OR–CI 95%	Gender	Age	Marital Status ^	Occupation *			Care Unit **				Children	Years of Service	EMPL ***
	F	Over 40	Married/ Cohabitant	Physicians	Nurses	Residents	Gen. ICU	Cardiac	Neon.	Ped.	Yes	Over 10	Public/Univ
10_yes	1.65 (0.96–2.84)	0.61 (0.35–1.08)		6.81 (1.70–27.20)	3.11 (0.89–10.93)	9.38 (2.07–42.41)	1.87 (1.07–3.29)	2.20 (0.89–5.39)					0.42 (0.14–1.22)
12_agree	0.54 (0.28–1.03)	2.63 (1.22–5.65)	2.02 (0.98–4.16)								0.36 (0.15–0.83)		
13_agree	2.13 (1.12–4.07)			5.01 (1.45–17.37)		10.71 (2.41–47.61)				4.64 (1.05–20.59)		1.75 (0.92–3.33)	
14_agree		0.37 (0.18–0.79)										2.16 (1.02–4.55)	
15_agree				6.46 (1.45–28.71)	5.15 (1.35–19.67)	11.49 (2.55–51.84)				3.08 (1.14–8.33)			0.29 (0.08–1.09)
16_agree		0.23 (0.06–0.84)											
17_agree									\	2.36 (0.94–5.97)		0.32 (0.19–0.55)	
18_agree		0.39 (0.17–0.90)		3.56 (0.80–15.90)					4.88 (1.41–16.85)				
19_agree	0.34 (0.11–1.10)				2.43 (0.86–6.84)			0.26 (0.08–0.88)	4.49 (0.97–20.79)				
20_agree											2.43 (1.21–4.87)	0.25 (0.12–0.49)	
21_yes					2.66 (1.07–6.58)	3.93 (1.45–10.64)		0.14 (0.02–1.12)				0.13 (0.26–1.08)	
25_yes		4.71 (2.84–7.79)	1.62 (0.98–2.69)							0.48 (0.22–1.05)			
26_agree		0.22 (0.11–0.45)						0.19 (0.08–0.49)					

^ reference: unmarried; * reference: Infectious Disease; ** reference: other health workers; *** reference: private.

**Table 5 vaccines-11-00196-t005:** Results of multivariate analysis—OR (95%CI) positive answers—chickenpox.

Outcome Variable OR–CI 95%	Gender	Age	Marital Status ^	Occupation *			Care Unit **				Children	Years of Service	Empl ***
	F	Over 40	Married/ Cohabitant	Physicians	Nurses	Residents	Gen. ICU	Cardiac	Neon.	Ped.	Yes	Over 10	Public/Univ
10_yes	1.70 (1.02–2.82)	0.44 (0.27–0.74)				2.25 (1.08–4.66)							
12_agree	0.37 (0.19–0.71)	2.79 (1.44–5.40)		0.37 (0.13–1.04)									
13_agree						4.43 (1.26–15.63)				4.92 (1.14–21.35)	0.42 (0.19–0.92)	2.31 (1.04–5.13)	
14_agree		0.49 (0.29–0.80)				0.51 (0.26–0.98)							
15_agree				10.78 (1.88–61.82)	10.45 (2.11–51.74)	20.58 (3.47–121.97)		0.40 (0.17–0.95)		4.55 (1.04–19.93)			0.18 (0.03–0.10)
16_agree		0.34 (0.15–0.80)		3.09 (0.69–13.84)									
17_agree										2.36 (0.94–5.97)		0.32 (0.19–0.55)	
18_agree				3.32 (0.74–14.79)					5.48 (1.58–18.96)			0.46 (0.20–1.04)	
19_agree	0.35 (0.11–1.10)				2.43 (0.86–6.84)			0.26 (0.08–0.88)	4.45 (0.97–20.79)				
20_agree				5.08 (1.06–24.37)	5.34 (1.27–22.49)	6.52 (1.36–31.32)		0.41 (0.18–0.97)					0.15 (0.03–0.69)
21_yes	0.51 (0.24–1.07)				2.91 (0.99–8.55)	5.67 (1.79–17.96)		0.16 (0.02–1.34)				0.17 (0.06–0.43)	
25_yes				1.91 (0.77–4.72)									
26_agree		0.41 (0.20–0.82)				3.32 (0.89–12.31)	0.49 (0.24–1.01)	0.17 (0.07–0.45)					

^ reference: unmarried; * reference: Infectious Disease; ** reference: other health workers; *** reference: private.

**Table 6 vaccines-11-00196-t006:** Results of multivariate analysis—OR (95%CI) positive answers—pertussis.

Outcome Variable OR–CI 95%	Gender	Age	Marital Status ^	Occupation *			Care Unit **				Children	Years of Service	Empl ***
	F	Over 40	Married/ Cohabitant	Physicians	Nurses	Residents	Gen. ICU	Cardiac	Neon.	Ped.	Yes	Over 10	Public/Univ
10_yes				3.02 (1.09–8.37)	2.38 (0.98–5.78)	7.02 (2.44–20.20)						0.45 (0.29–0.82)	
12_agree				2.73 (1.15–6.50)	2.39 (1.19–4.77)								
13_agree	2.32 (1.23–4.39)			8.51 (1.95–37.07)		8.68 (2.02–37.23)				2.71 (0.78–9.41)			
14_agree		0.44 (0.21–0.93)										2.11 (1.01–4.39)	
15_agree				7.12 (1.57–32.19)	4.60 (1.19–17.77)	12.75 (2.79–58.35)	0.61 (0.35–1.07)	0.46 (0.19–1.11)		4.62 (1.32–16.12)			0.28 (0.07–1.10)
16_agree		0.46 (0.19–1.15)					0.42 (0.17–1.06)						
17_agree			1.92 (1.08–3.41)	4.19 (1.56–11.25)						2.34 (0.90–6.05)		0.31 (0.17–0.54)	
18_agree		0.38 (0.16–0.90)		3.53 (0.79–15.87)		5.88 (0.72–48.28)			7.93 (1.82–34.57)				
19_agree	0.25 (0.07–0.88)				2.64 (0.94–7.44)			0.23 (0.07–0.79)	2.83 (0.76–10.56)				
20_agree				4.68 (0.10–22.02)	4.01 (0.98–16.38)	3.89 (0.76–19.85)						0.58 (0.31–1.09)	0.19 (0.04–0.88)
21_yes						3.12 (1.50–6.47)						0.11 (0.04–0.33)	
25_yes	1.83 (0.96–3.52)	2.86 (1.62–5.04)			0.61 (0.35–1.09)								
26_agree		0.43 (0.17–1.08)					0.38 (0.17–0.82)	0.19 (0.08–0.50)				0.40 (0.15–1.06)	

^ reference: unmarried; * reference: Infectious Disease; ** reference: other health workers; *** reference: private.

**Table 7 vaccines-11-00196-t007:** Results of multivariate analysis—OR (95%CI) positive answers—meningococcal disease.

Outcome Variable OR–CI 95%	Gender	Age	Marital Status ^	Occupation *			Care Unit **				Children	Years of Service	Empl ***
	F	Over 40	Married/ Cohabitant	Physicians	Nurses	Residents	Gen. ICU	Cardiac	Neon.	Ped.	Yes	Over 10	Public/Univ
10_yes	1.98 (1.17–3.33)	0.36 (0.22–0.60)		2.59 (1.28–5.26)				3.01 (1.17–7.76)					
12_agree	0.37 (0.17–0.80)	2.2 (0.98–4.94)								2.47 (0.94–6.51)			
13_agree	2.72 (1.333–5.54)		2.28 (0.96–5.41)			9.24 (2.02–42.19)					0.37 (0.13–1.05)		
14_agree		0.42 (0.20–0.87)										1.87 (0.90–3.88)	
15_agree				6.31 (2.01–19.80)	4.68 (2.01–10.90)	5.26 (1.84–15.07)		0.45 (0.18–1.09)		3.32 (0.95–11.60)			
16_agree				14.29 (1.64–124.23)	2.78 (1.04–7.43)	17.32 (1.10–150.17)				4.39 (0.57–33.95)			
17_agree							2.15 (0.89–5.10)		2.50 (1.04–5.99)			0.53 (0.25–1.12)	
18_agree		0.35 (0.12–0.02)											
19_agree						4.19 (0.55–31.98)							
20_agree				7.08 (1.39–36.00)	6.73 (1.55–29.19)	6.94 (1.40–34.39)		0.38 (0.16–0.92)					0.16 (0.03–0.80)
21_yes	0.49 (0.26–0.92)			7.18 (1.40–36.86)	6.47 (1.38–30.28)	14.9 (2.10–74.11)	0.51 (0.27–0.96)	0.12 (0.02–0.62)				0.23 (0.12–0.47)	
25_yes	0.14 (0.02–0.87)								5.36 (0.96–29.87)				
26_agree		0.23 (0.11–0.48)						0.22 (0.09–0.54)					

^ reference; unmarried; * reference: Infectious Disease, ** reference: other health workers; *** reference: private.

**Table 8 vaccines-11-00196-t008:** Results of multivariate analysis—OR (95%CI) positive answers—source of information.

Outcome Variable OR–CI 95%	Gender	Age	Marital Status ^	Occupation *			Care Unit **				Children	Years of Service	Empl ***
	F	Over 40	Married/ Cohabitant	Physicians	Nurses	Residents	Gen. ICU	Cardiac	Neon.	Ped.	Yes	Over 10	Public/Univ
colleagues_11_yes		0.64 (0.40–1.02)			0.63 (0.39–1.02)			2.55 (1.03–6.28)	1.72 (0.97–3.03)	2.76 (1.28–5.92)			
societies_11_yes		0.41 (0.23–0.74)	1.74 (1.02–2.95)		5.16 (1.66–16.05)	14.35 (3.92–52.48)							
journals_11_yes		0.41 (0.23–0.76)		88.81 (18.97–415.72)	5.46 (1.57–19.00)	7.73 (1.99–30.04)	0.54 (0.30–0.96)						
physician_11_yes					3.98 (2.08–7.65)	1.96 (0.89–4.32)	0.52 (0.31–0.90)			3.76 (1.31–10.43)			0.24 (0.09–0.64)
mass_11_yes				0.30 (0.12–0.74)		0.13 (0.04–0.43)			1.66 (0.92–3.00)				
social_11_yes				0.15 (0.04–0.67)		0.16 (0.05–0.55)					0.48 (0.25–0.92)		
websites_11_yes		0.41 (0.21–0.81)		4.15 (1.42–12.10)	3.58 (1.53–8.36)	7.75 (1.81–33.27)				0.47 (0.21–1.07)			

^ reference: unmarried; * reference: Infectious Disease; ** reference: other health workers; *** reference: private.

## Data Availability

Data supporting reported results can requested to the authors.

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
