# Peer review of "Analysis of Knowledge, Attitudes and Behaviours of Health Care Workers towards Vaccine-Preventable Diseases and Recommended Vaccinations: An Observational Study in a Teaching Hospital"

_vaccines, 2023, doi:10.3390/vaccines11010196_

Round 1

Reviewer 1 Report

The paper reports an interesting research on vaccination perception in health care workers. The study is interesting and deserve publication.

However, I have some suggestions to improve the paper.

 1. In abstract please insert OR together with CI95% in the appropriate position to help readers to understand

2. M&M please report references and validation for the questionnaire used

3.   Line 124= p<0.05 I think

4. Years of service and years of service CU: the groups formed are not well distributed with more than 40% in the first group. I think that an analysis that considered age classes in better divided groups will permit better results (i.e. divide age classes by quartiles)

5. Please report the number or responders on the total population involved in the study

6. Please insert a title for figure 1

7. Please insert CI95% after OR reported in the txt

8. Table 2: this table is difficult to understand. Tables need to be self-explanatory. I suggest to improve the title and report the meaning of measle 12 -14 etc. May be would be useful to rotate the table, to avoid the repetition of “measle” and to report the meaning of the numbers in legend or in M&M section

9.Same observation for the other tables following

10.   Same observation for tables supplemental. Please rotate the table to increase readability

11.  Line 303 … Nurses I suggest “to be a nurse …”

12.  Discussion “they were less accepting of mandatory vaccinations and future booster vaccinations, if necessary”. This sentence is contrast with results section “Most of the participants were in favour of mandatory vaccination, with the highest agreement seen for meningococcal disease (81.5%), followed by chickenpox (77.6%), measles (76.6%), pertussis (76%), rubella (75.3%) and mumps (75%).”

Reviewer 2 Report

In my opinion the paper is informative and provides a valuable source document for anyone requiring a primer to know and understand this issue.  This article reports on original primary research.     Overall, the article follows an appropriate structure.   The Introduction of the study is well described. The knowledge gap is outlined well for the most part.  The aim of this study is well stated and of interest.   But, some shortcomings in the sections Methods make this paper not appropriate for publication in this form and certain corrections should be made (minor revision).

Some comments:        

- Lines 79-89: Describe in detail: were all units in this hospital selected, if not - state according to what criteria were exactly those units chosen, study population (total number of employees, etc.), study sample calculation (with an appropriate reference), state the exclusion criteria, state whether all HCWs gave a written consent to participate in the study.

- Line 90-103: At the end of the paper add the whole questionnaire. It might help the readers get a more clear picture of all results.

- Line 104-125: Was collinearity between the variables that were included in the model of logistic regression analysis examined, inscribe the results that were obtained for this. State what level of probability was used to enter the variables in the model of multivariate logistic regression analysis.  

- Lines 422-429: Some data (e.g. on immunity) were self-reported, without an insight into relevant documentation or laboratory confirmation, which can be a study limitation. Especially since not all responders were health care providers.  The discussion focused on the comparison of own results with the results of other studies.    

The study's limitations are discussed.   

Conclusions are overall supported by the results.    
